# Ongoing evolution of the *Mycobacterium tuberculosis* lactate dehydrogenase reveals the pleiotropic effects of bacterial adaption to host pressure

**Sydney Stanley**[1], **Xin Wang**[1], **Qingyun Liu**[1], **Young Yon Kwon**[2], **Abigail M. Frey**[1], **Nathan D. Hicks**[1], **Andrew J. Vickers**[1], **Sheng Hui**[2], **Sarah M. Fortune**[1,3,4]*

1 Department of Immunology and Infectious Diseases, Harvard T.H. Chan School of Public Health, Boston, Massachusetts, United States of America, 2 Department of Molecular Metabolism, Harvard T. H. Chan School of Public Health, Boston, Massachusetts, United States of America, 3 Ragon Institute of MGH, MIT, and Harvard, Cambridge, Massachusetts, United States of America, 4 Broad Institute of MIT and Harvard, Cambridge, Massachusetts, United States of America

* sfortune@hsph.harvard.edu

**Data Availability Statement:** All data are available in the main text or supplementary materials;

## Abstract

The bacterial determinants that facilitate *Mycobacterium tuberculosis* (Mtb) adaptation to the human host environment are poorly characterized. We have sought to decipher the pressures facing the bacterium *in vivo* by assessing Mtb genes that are under positive selection in clinical isolates. One of the strongest targets of selection in the Mtb genome is *lldD2*, which encodes a quinone-dependent L-lactate dehydrogenase (LldD2) that catalyzes the oxidation of lactate to pyruvate. Lactate accumulation is a salient feature of the intracellular environment during infection and *lldD2* is essential for Mtb growth in macrophages. We determined the extent of *lldD2* variation across a set of global clinical isolates and defined how prevalent mutations modulate Mtb fitness. We show the stepwise nature of *lldD2* evolution that occurs as a result of ongoing *lldD2* selection in the background of ancestral lineage-defining mutations and demonstrate that the genetic evolution of *lldD2* additively augments Mtb growth in lactate. Using quinone-dependent antibiotic susceptibility as a functional reporter, we also find that the evolved *lldD2* mutations functionally increase the quinone-dependent activity of LldD2. Using $^{13}$C-lactate metabolic flux tracing, we find that *lldD2* is necessary for robust incorporation of lactate into central carbon metabolism. In the absence of *lldD2*, label preferentially accumulates in dihydroxyacetone phosphate (DHAP) and glyceraldehyde-3-phosphate (G3P) and is associated with a discernible growth defect, providing experimental evidence for accrued lactate toxicity via the deleterious buildup of sugar phosphates. The evolved *lldD2* variants increase lactate incorporation to pyruvate while altering triose phosphate flux, suggesting both an anaplerotic and detoxification benefit to *lldD2* evolution. We further show that the mycobacterial cell is transcriptionally sensitive to the changes associated with altered *lldD2* activity which affect the expression of genes involved in cell wall lipid metabolism and the ESX-1 virulence system. Together, these data illustrate a multifunctional role of LldD2 that provides context for the selective advantage of *lldD2* mutations in adapting to host stress.

sequencing data are deposited in the Sequence Read Archive database (PRJNA1026166).

**Funding:** National Institutes of Allergy and Infectious Diseases (https://www.niaid.nih.gov): P01 AI132130 (S.S., S.M.F.); P01 AI143575 (X.W., S.M.F.); U19 AI142793 (Q.L., S.M.F.); 5T32AI132120-03 (S.S); 5T32AI132120-04 (S.S.); 5T32AI049928-17 (S.S.); Harvard Graduate Program in Tropical Infectious Diseases Training Grant T32AI049928 (A.M.F.) National Institute of Child Health and Human Development (https://www.nichd.nih.gov): 5T32HD040128-19 (S.S.) National Institutes of Health (https://www.nih.gov): R00DK117066 (S.H.) The funders had no role in study design, data collection and analysis, decision to publish, or preparation of the manuscript.

**Competing interests:** The authors have declared that no competing interests exist.

## Author summary

Mycobacterium tuberculosis (Mtb) is the causative agent of the deadly disease tuberculosis (TB). Mtb has infected humans for thousands of years, and over the millennia has adapted to circumvent the challenges presented by the human host. Characterizing how Mtb has evolved to contend with host pressure can improve our understanding of mechanisms that modulate bacterial fitness, which we can ultimately leverage to design better antitubercular agents and diagnostics. One way to assess ways in which Mtb is adapting to host stress is to characterize bacterial genes that are diversifying in patients. Here we focus on the gene *lldD2*, and we illustrate the stepwise acquisition of *lldD2* mutations throughout Mtb evolutionary history. LldD2 mediates the metabolism of lactate, which is an abundant carbon source during infection. We show that both ancient and contemporary *lldD2* mutations increase the ability of Mtb to grow in the presence of lactate, and these mutations have an additive effect. In addition, *lldD2* mutations mitigate potential toxicity caused by lactate metabolism and also modulate the expression of genes that are involved in pathogenesis. In all, our data suggests that lactate is a long-standing host pressure and that Mtb adaption has ranging effects that augment bacterial fitness.

## Introduction

*Mycobacterium tuberculosis* (Mtb) is currently the leading infectious cause of death globally [1]. The temporal and geographic extent of the tuberculosis (TB) pandemic reflects remarkable adaption to human populations [2,3]. Mtb is distinguished by limited genetic diversity due to a low mutation rate and minimal horizontal gene transfer and recombination events [4–6]. However, the seven genetically divergent human-adapted lineages of the *Mycobacterium tuberculosis* complex (MTBC) have distinct epidemiologic characteristics [7], suggesting that the mutations that define these lineages and sub-lineages are functionally important. Mtb strains also continue to acquire mutations that may be selectively advantageous in the face of the host environment and/or antibiotic pressures [8–11]. Studies defining the phenotypic consequences of positive selection as a result of drug pressure have uncovered altered antibiotic sensitivity phenotypes conferring resistance, tolerance, and resilience [8,12–19]. Considerably less is known about the functional consequence of positive selection driven by host pressure or the impact of lineage and sub-lineage defining mutations on host fitness.

In the current study, we sought to delineate the impact of functional genetic diversity in response to host challenge. We focus here on *lldD2* (*rv1872c*), an Mtb gene that encodes the lactate dehydrogenase (LDH) LldD2, which has been shown to be required for Mtb grown on lactate as a sole carbon source [20]. LldD2 is a quinone-dependent LDH and employs flavin mononucleotide as an electron carrier for the oxidation of lactate [20]. Previous studies indicate that *lldD2* has one of the strongest signatures of nonantibiotic-related diversifying selection and mutations are prevalent in clinical strains [8,10]. There are lineage and sub-lineage defining mutations in *lldD2* which were acquired thousands of years ago [10], and mutations that continue to emerge contemporaneously in clinical strains from different branches of the phylogenetic tree [8,10,21]. In addition, *lldD2* is under positive selection in *M. kansasii*, an environmental nontuberculous mycobacterium with an increasing incidence of human infection, which may suggest that *lldD2* is important for mycobacterial adaption to the human host [22].

Lactate is a prominent feature of the host environment; it is a significant contributor to tricarboxylic acid (TCA) cycle metabolism across several tissue types, especially the lung [23].

Further, lactate can be considered an immune-regulated host feature. Mtb infection in mice and IFNγ-activated macrophages induces the Warburg effect, in which host lactate is produced as a result of aerobic glycolysis [24,25]. Furthermore, lactate accumulation has been identified in mouse and guinea pig lung granulomas [26,27]. Deletion of *lldD2* impairs Mtb survival in macrophages, demonstrating a requirement for lactate metabolism *in vivo* [20]. Consistent with the *in vivo* relevance, a *lldD2* promoter mutation (-18 G>T) that is prevalent among clinical Mtb strains is associated with increased transmissibility [28].

It is unclear why *lldD2* is necessary for Mtb infection. In this study, we assess the functional effects of *lldD2* evolution. Utilizing a panel of Mtb clinical isolates and isogenic strains, we demonstrate that selected *lldD2* variants augment Mtb growth in lactate, modulate metabolic flux, and have striking secondary effects on expression of critical virulence systems. Taken together, these data inform our understanding of the evolution of Mtb to the human host and critical processes in Mtb pathogenesis.

## Results

### Stepwise evolution of *lldD2* increases Mtb fitness in lactate

A previous study analyzed genome sequences from 220 MTBC isolates and characterized multiple genotypes of *lldD2* [10]. They identified that three *lldD2* codons (3, 109, and 253) were subject to diversifying selection, and determined that a V253M mutation had been acquired by the ancestor of L2 before lineage diversification occurred [10]. Subsequently, our work and that of others, which involved much larger datasets of MTBC whole-genome sequences, have revealed that *lldD2* is under positive selection, and this selection process is still ongoing in contemporary MTBC strains [8,28]. Together, these observations suggest that selective pressure on *lldD2* has persisted since the very beginning of MTBC diversification, enduring over thousands of years.

To gain deeper insights into the evolutionary dynamics of the *lldD2* gene within the MTBC population, we assembled the genome sequences of 296 MTBC strains, which collectively represented the major lineages and sub-lineages and included both drug-resistant and sensitive strains [29]. Employing the *lldD2* sequence derived from the inferred most recent common ancestor of the MTBC, we reconstructed the gene's evolutionary history within the MTBC population. We find that there are root mutations in *lldD2* in every Mtb lineage except L1 and L6 (Fig 1). The modern Mtb lineages (L2, L3, and L4) and L7 share an ancestral A176V mutation and L4 shares an additional lineage-defining A59G mutation (Fig 1). L5 strains, which are highly prevalent in West Africa, derived an A2A and A237S from their common ancestor (Fig 1). Notably, we also observed a remarkable pattern of stepwise evolution in *lldD2*, resulting in the accumulation of up to four distinct mutations in some sub-lineages or clades (Fig 1). For instance, following the acquisition of the A176V and A59G mutations, a specific clade within L4 further accumulated V253M and L96F mutations (Fig 1). Derived mutations span the *lldD2* promoter and coding regions but the V3I, V253M, and the -18 G>T promoter mutations in particular have recurrently evolved independently among MTBC strains (Figs 1 and 2A) [10,21,28]. In an analysis of over 50,000 global MTBC clinical strains encompassing lineages L1-L7, these three mutations account for 52% of the 17,000 *lldD2* homoplastic or clade-defining mutations occurring across 216 different loci (Fig 2B). The convergent signals of these homoplastic mutations strongly imply their adaptive role during the evolution of MTBC [8,10,28]. The stepwise accumulation of multiple adaptive mutations suggests the existence of a complex and dynamic fitness landscape governing the evolution of *lldD2*. In this landscape, a single ancestral lineage-defining mutation may enhance fitness, while specific additional mutations can drive fitness even higher, thus indicating the intricate nature of LldD2's evolutionary trajectory.

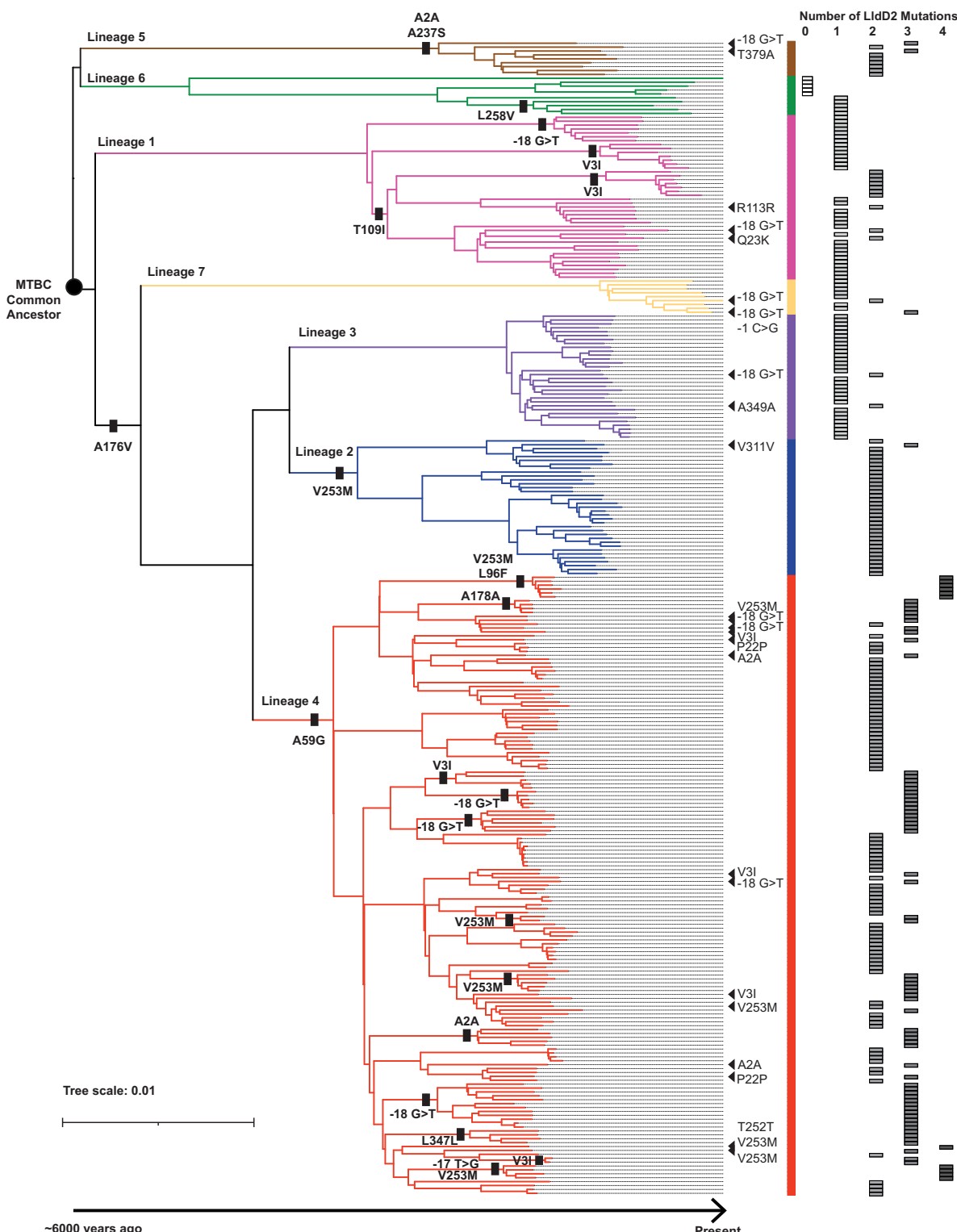

**Fig 1. Stepwise evolution of LldD2.** Phylogenetic tree of 296 *M. tuberculosis* and *M. africanum* clinical isolates representing the global diversity of the *Mycobacterium tuberculosis* complex (MTBC). The tree scale denotes number of mutations per site. Ancestrally derived lineage and sub-lineage defining *lldD2* mutations are indicated by a bar. Triangles denote *de novo* or homoplastic mutations. Total number of mutations per strain is tallied to the right of the phylogenetic tree.

The stepwise evolution of *lldD2* is especially striking in comparison to *lldD1*, which does not have LDH activity but is thought to participate in the oxidation of other small α-hydroxy acids [20]. *LldD1* mutations have accumulated in only 10% of the 50,000 MTBC strains, and most of these variants are concentrated in L1 (Fig 2C–2D). However, 99.6% of MTBC strains carry at least one *lldD2* mutation (Fig 2C). This distinct pattern of selection over long time scales, which is distinct from the much more recent patterns of positive selection on known drug resistance genes [8], strongly suggests that MTBC strains are evolving to optimize lactate metabolism as a result of immune-metabolic pressures it encounters in the human host.

To define the functional effects of the *lldD2* variants, we first identified drug-sensitive clinical strains of Mtb with various *lldD2* mutations from a panel of strains isolated from TB patients in Ho Chi Minh City, Vietnam [30] (S1 Table). The clinical isolates belong to L1, L2, and L4, which are globally the three most prevalent lineages [31]. We focused on the -18 G>T promoter mutation, V3I, and V253M because we identified both lineage/clade-defining and homoplastic occurrences of these variants in clinical strains (Fig 1). We first assessed the growth of strains carrying the L2 ancestral allele versus V3I + M219L mutations in standard media (7H9), which contains 0.2% glycerol as a carbon source but also a supplement including oleic acid and dextrose, and 7H9 media with the addition of 10 mM L-lactate. In the presence of lactate, the clinical isolate carrying the derived variants had increased growth compared to the strain carrying the ancestral allele (S1 Fig). We then expanded this analysis, testing the full panel of strains with derived and ancestral alleles in 7H9 and defined media with lactate as the sole carbon source (7H12–0.2% L-lactate). We utilized single carbon source media so that we could more precisely attribute the phenotypes of *lldD2* variants to lactate. After normalizing to growth in 7H9, we find that clinical isolates that carry the evolved *lldD2* alleles have increased growth in lactate media as compared to nearest-neighbor strains that only carry ancestral *lldD2* alleles (unpaired t-test, P = 0.037) (Fig 3A–3B).

We next constructed an *lldD2* deletion mutant in H37Rv, the L4 derived lab-adapted strain. We complemented the mutant with a panel of *lldD2* alleles that recapitulate the evolution of *lldD2* in L4 from the inferred common ancestor sequence to the homoplastic variants present in L4 clinical strains (Figs 1 and 3C). In the engineered strains, expression of *lldD2* is driven by a constitutive promotor so the -18 G>T mutation was not assessed. In multiple carbon source media (7H9), all of the strains demonstrate similar growth patterns (Figs 3D and S2). In lactate media, Δ*lldD2* and Δ*lldD2* complemented with a vector control (Δ*lldD2*::Control) do not grow appreciably. However, the addition of each successive *lldD2* mutation measurably augments Mtb growth in lactate, except for A59G for which we did not find a fitness benefit (Figs 3D and S2).

Finally, we constructed isogenic allelic variants by introducing the point mutations within the native locus of *lldD2*. We constructed these variants in H37Rv, which contains the homoplastic -18 G>T promoter mutation (Fig 3E). Therefore, to create a strain expressing the L4 ancestral *lldD2* sequence, we introduced a -18 T>G mutation (Fig 3E). We also introduced the V3I and V253M mutations into wild-type H37Rv (Fig 3E). In 7H9, there were minimal quantitative differences in growth between strains (Figs 3F and S2). However, in lactate media, we again see the strains expressing the evolved mutations grow faster than the strain expressing the L4 ancestral allele (-18 T>G) (Figs 3F and S2).

## Homoplastic *lldD2* mutations alter gene and enzyme activity

To determine how the homoplastic *lldD2* mutations result in gain-of-function activity, we first assessed the effect of the promoter and early codon mutations on *lldD2* expression. We found that the -18 G>T promoter mutation increases *lldD2* expression relative to the ancestral allele

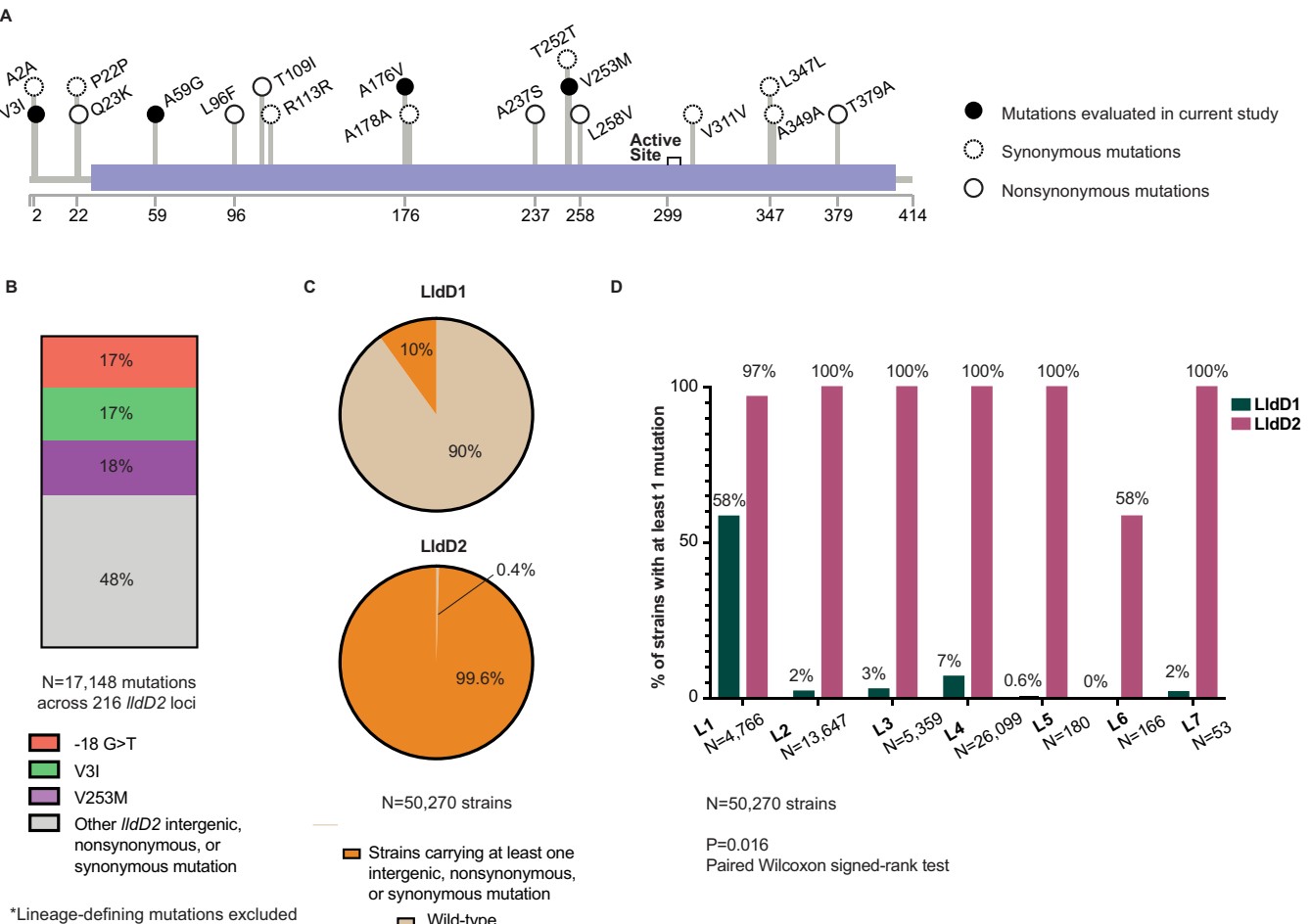

**Fig 2. Prevalence of various LldD2 mutations.** (A) Schematic of LldD2 depicting the amino acid position of coding mutations listed in Fig 1. The purple rectangle illustrates the FMN-dependent alpha-hydroxy acid dehydrogenase motif. (B) Percentage of *lldD2* homoplastic or clade-defining mutations belonging to the indicated genotype. 17,148 mutations spanning 216 unique loci were identified across 50,270 strains. (C) Percentage of strains carrying at least one of the indicated mutations in either LldD1 or LldD2. (D) Comparison of the percentage of strains carrying at least one LldD1 or LldD2 mutation for each lineage.

(ordinary one-way ANOVA, Dunnett's multiple comparison test, P = 0.011) (Fig 4A). The addition of the V3I mutation (ordinary one-way ANOVA, Dunnett's multiple comparison test, P = 0.025) but not the V253M mutation (ordinary one-way ANOVA, Dunnett's multiple comparison test, P = 0.74) to the -18 G>T background also results in increased *lldD2* expression. To determine whether this results in increased LldD2 protein production, we constructed *M. smegmatis* (Msm) strains that express the first 60 amino acids of LldD2 fused to Renilla luciferase, with expression driven by the *lldD2* native promoter. Therefore, we could quantify luminescence as a proxy for LldD2 production. In this system, the -18 G>T mutation alone results in a minimal increase in LldD2 production compared to the L4 ancestral promoter (ordinary one-way ANOVA, Dunnett's multiple comparison test, P = 0.48), but consistent with the gene expression data, both the V3I and the -18 G>T + V3I variants significantly increase protein production (ordinary one-way ANOVA, Dunnett's multiple comparison test, P = 0.0002 and P<0.0001 respectively) (Fig 4B).

LldD2 mediates the oxidation of lactate into pyruvate, which is a quinone-dependent reaction [20]. Given that we have thus far demonstrated that the homoplastic *lldD2* mutations are gain-of-function, we hypothesized that the variants might affect Mtb susceptibility to

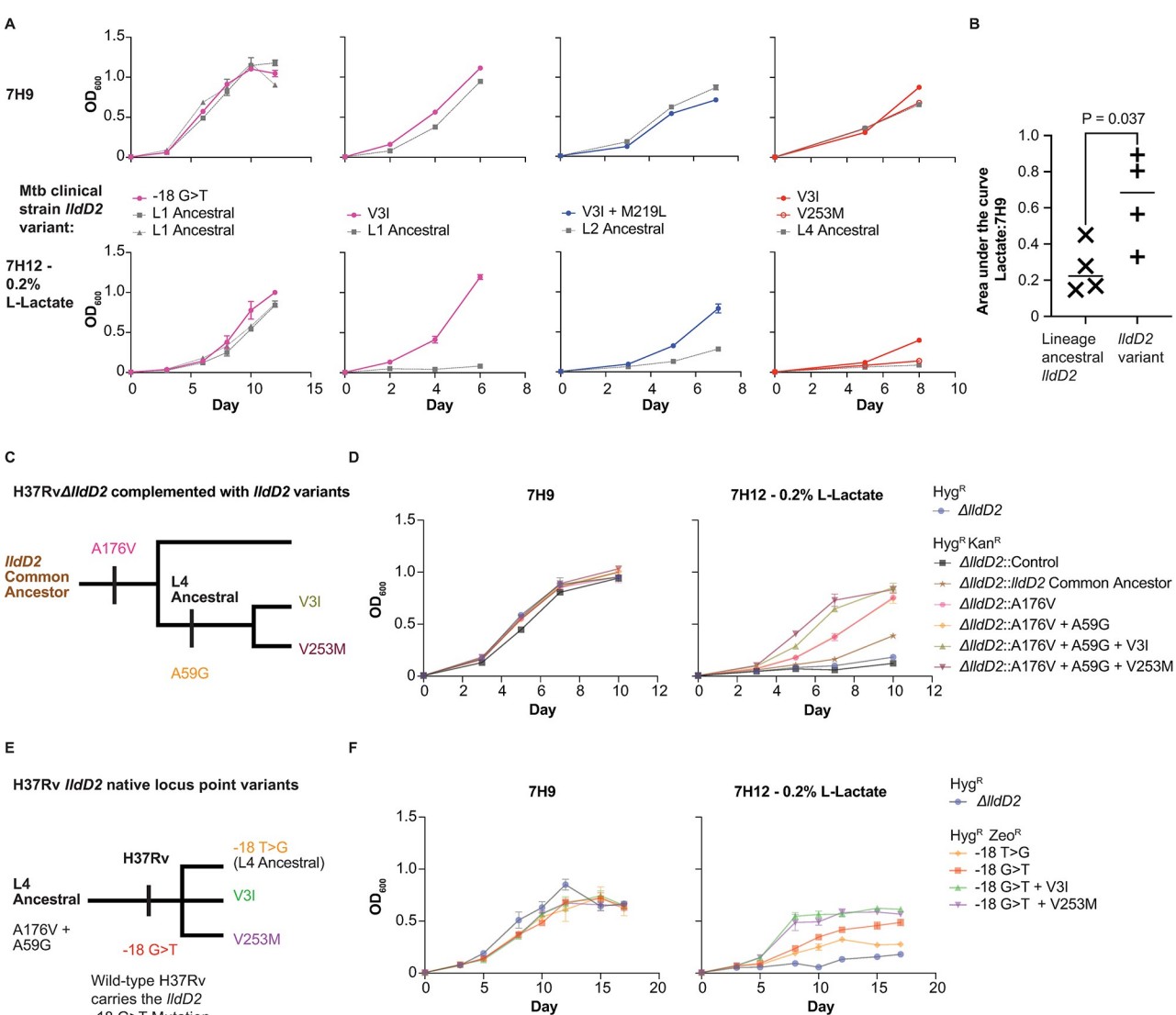

**Fig 3. Ancestral and homoplastic mutations in *lldD2* augment Mtb fitness in lactate.** (A) Growth curves of Mtb clinical strains from L1, L2, and L4. Strains are grouped for comparison of the *lldD2* homoplastic mutation strains to closely related lineage ancestral strains. All cultures started at $OD_{600}$ 0.005 at day 0. Triplicate replicates shown, error bars represent the standard deviation. (B) Quantification of the area under the curves (AUCs) from A. The AUC in 7H12–0.2% L-lactate is normalized to that of 7H9 for a given strain. The AUC is averaged for the V3I and V253M L4 strains and for the two L1 ancestral strains that were paired to -18 G>T. P-value indicates results of an unpaired t-test. (C) and (E) Schematic depicting the construction of recombinant *lldD2* strains. (D) and (F) Growth curves of the strains indicated in (C) and (E). Hyg$^R$, Kan$^R$, and Zeo$^R$ refer to hygromycin, kanamycin, and zeocin resistance respectively. All cultures started at $OD_{600}$ 0.005 at day 0. Triplicate replicates shown, error bars represent the standard deviation. Representative of two independent experiments.

clofazimine (CFZ). CFZ binds to NDH-2, the main NADH:quinone reductase of the Mtb electron transport chain, with higher binding affinity than menaquinone; oxidation of CFZ results in reactive oxygen species (ROS) production which ultimately leads to Mtb killing [32]. Accordingly, CFZ sensitivity can serve as a probe of quinone-dependent enzyme activity. We quantified CFZ susceptibility with a growth-based Alamar blue reduction assay. In 7H9, all strains exhibit the same sensitivity to CFZ (Fig 4C). In media with lactate as the sole carbon source, Mtb strains expressing homoplastic *lldD2* variants were significantly more sensitive to CFZ compared to the -18 T>G strain expressing the L4 ancestral allele at concentrations of

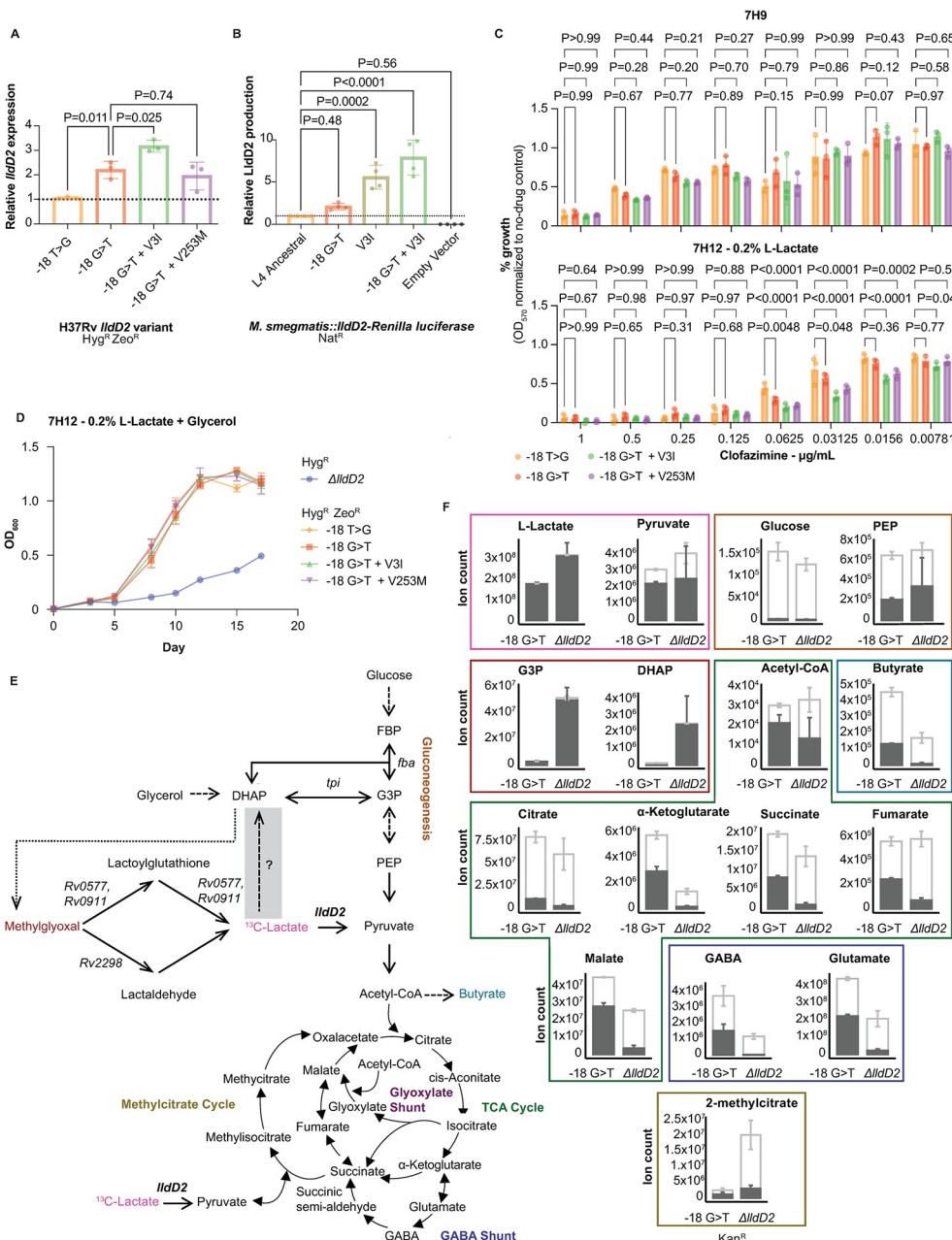

**Fig 4. *lldD2* homoplastic mutations modulate gene and enzyme activity.** (A) *lldD2* expression as measured by qPCR. Gene expression is normalized to that of -18 T>G. Each dot represents average of technical replicates from one of three independent experiments. Error bars indicate standard deviation. P-value indicates results of an ordinary one-way ANOVA test with Dunnett's multiple comparison correction. (B) LldD2 production as measured by a Renilla luciferase assay. Luminescence normalized to that of the Msm strain expressing the L4 ancestral version of *lldD2*. Each dot represents average of technical replicates from an independent experiment, error bars represent the standard deviation. P-value indicates results of an ordinary one-way ANOVA test with Dunnett's multiple comparison correction. Nat[R] refers to nourseothricin resistance. (C) Alamar blue assay of clofazimine (CFZ). P-value indicates the results of two-way ANOVA with Dunnett's multiple test correction. Triplicate replicates shown, error bars represent the standard deviation. (D) Growth curves of the indicated strains. All cultures started at OD$_{600}$ 0.005 at day 0. Triplicate replicates shown, error bars represent the standard deviation. Representative of two independent experiments. (E) Schematic of $^{13}$C-Lactate metabolic flux assay. Grey box indicates a proposed pathway. Dotted arrow represents the spontaneous conversion of DHAP to methylglyoxal. Dashed arrows represent abbreviated steps. (F) $^{13}$C-Lactate metabolic flux assay results. Error bars represent the standard deviation of three technical replicates. Dark grey indicates labeled carbon ions, light grey represents total carbon ions.

CFZ ranging from 0.0156–0.0625 μg/ml; the strains with two derived alleles showed greater sensitivity than the strain with the -18 G>T mutation alone (Fig 4C). All strains demonstrated the same susceptibility to trifluoperazine, which targets NDH-2 but does not function as a quinone competitor [33] (S3 Fig). These data suggest that *lldD2* gain-of-function variants additively augments quinone cycling during lactate metabolism.

Billig *et al*. showed that *lldD2* is necessary for growth with lactate as the sole carbon substrate [20]. To examine *lldD2* essentiality in the context of additional carbon sources, we completed growth curves in 7H12 with 0.2% lactate and glycerol. The *ΔlldD2* strain demonstrated markedly slower growth compared to the isogenic variants in lactate and glycerol (Figs 4D and S2) but the *ΔlldD2* strain did not exhibit a growth defect with glycerol as the sole carbon source (S4 Fig). This implies that growth in lactate is deleterious to Mtb without a functional lactate dehydrogenase, even in the presence of another carbon source. Thus, we postulated that LldD2 has a detoxification function in addition to its role in anaplerosis. We reasoned that impaired growth of the *ΔlldD2* strain may be due to the disruption of this detoxification process.

To test this model, we constructed a pair of *ΔlldD2* and -18 G>T strains in the same parental genetic backgrounds (S5 Fig) (S1 Table). We performed a metabolic flux assay after 8 hours of growth on $^{13}$C-lactate and glycerol and observed markedly more $^{13}$C-labeled dihydroxyacetone phosphate (DHAP) and glyceraldehyde-3-phosphate (G3P) in the *ΔlldD2* strain compared to the -18 G>T strain (Fig 4E–4F) (S2 Table). DHAP and G3P were nearly completely labeled in the *ΔlldD2* strain, indicating that these metabolites were the products of $^{13}$C-lactate metabolism. We also observed significant $^{13}$C incorporation of phosphoenolpyruvate (PEP) but not glucose, which is indicative of glycolytic but not gluconeogenic flux of lactate under this growth condition (Fig 4E–4F). We found labeled lactate and pyruvate in both -18 G>T and *Δlldd2* strains (Fig 4F). Together, these data indicate that without LldD2 activity, $^{13}$C-lactate buildup favors DHAP and G3P formation. This drives glycolytic flow to pyruvate. However, accumulation of these triose phosphate intermediates can be toxic [34], and this toxicity may contribute to the growth impairment of the *ΔlldD2* strain in the presence of lactate (Fig 4D). This data also implies a metabolic pathway exists in which lactate is a precursor of DHAP or G3P synthesis (Fig 4E).

We completed the same experiment with the Mtb strains expressing the evolved *lldD2* point mutants. We compared the ancestral -18 T>G strain to -18 G>T, -18 G>T + V3I, and -18 G>T + V253M variants and similarly found increased G3P labeling in the ancestral *lldD2* strain compared to those carrying evolved variants (S6 Fig). Together, the growth curve and metabolic flux data provide experimental evidence for a role of LldD2 in both incorporating lactate into central carbon metabolism and ameliorating potential lactate toxicity mediated by sugar phosphate accumulation. This pathway may be a contributing selective force driving gain-of-function *lldD2* diversification.

## Lactate rewires transcriptomic networks involving Mtb cell wall lipids and virulence systems

We next sought to identify downstream effects of *lldD2* variation during Mtb metabolism of lactate. We performed a genome-wide gene expression analysis of the -18 G>T and -18 T>G strains after 6-hour exposure to 7H9 or in 7H12 with lactate as the sole carbon source. A principal component analysis (PCA) of the technical replicates based on reads demonstrated that lactate drives significant differences in Mtb gene expression (Fig 5A). In 7H9, just seven genes are significantly upregulated (log2 fold change (L2FC) greater than 0.6 or less than -0.6 and P < 0.01) in the -18 G>T strain compared to the -18 T>G strain (Fig 5B and S3 Table). Of

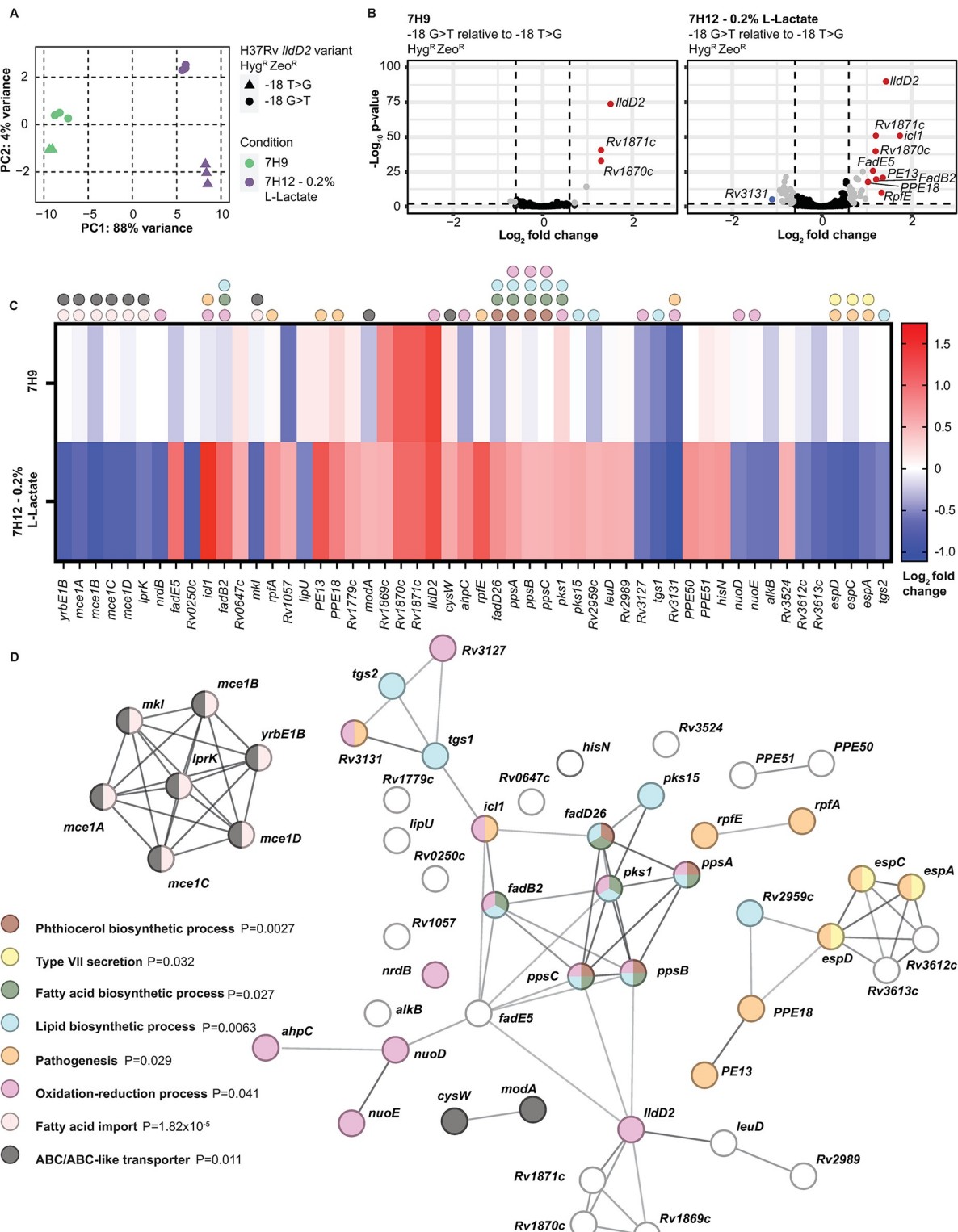

**Fig 5. Lactate and *lldD2* variants remodel the Mtb virulence transcriptome.** (A) Principal component analysis of the RNA-seq technical replicates based on normalized counts. (B) Volcano plot of the differential expression (DE) analysis. Grey indicates DE genes with log₂ fold change (L2FC) greater than 0.6 or less than -0.6 (P<0.01), red indicates DE genes with an L2FC greater than 1, blue indicates DE genes with an L2FC less than -1. There are 7 DE genes in 7H9 and 52 DE genes in lactate. (C) Heatmap of the 52 DE genes from the lactate condition, 7H9 shown for comparison. The genes are ordered according to chromosome position. Dots denote genes involved in the pathways shown

in (D). (D) STRING protein-protein interaction network of the 52 DE genes from the lactate condition. P-value indicates the significance of Gene Ontology, KEGG, or STRING pathway enrichment after multiple test correction.

these, only three genes have L2FC >1: *lldD2*, Rv1871c, and Rv1870c, consistent with our directed analysis (Fig 3A) and operonic expression of Rv1872c-Rv1870c (Fig 5B and S3 Table).

In the lactate growth condition, 52 genes were differentially expressed (L2FC greater than 0.6 or less than -0.6, P<0.01) comparing the derived -18 G>T to the ancestral -18 T>G (Fig 5B–5C) (S3 Table). We utilized this gene set to create a protein-protein interaction network based on the STRING database [35] and performed functional enrichment analysis according to Gene Ontology biological processes [36], STRING network clusters, and KEGG pathways [37]. This gene set has significantly more interactions compared to a sample randomly selected from the genome (P = 2.21x10$^{-11}$), which is indicative of a coordinated biological response [35]. During lactate metabolism, the -18 G>T mutation modulates expression of gene programs largely involved in fatty acid metabolism and virulence. The gene set is enriched for genes that coordinate fatty acid import (P = 1.82x10$^{-5}$), notably the *mce1* operon, which is downregulated in the strain with the evolved allele (Fig 5C–5D). Differently expressed genes are also enriched for mediators of fatty acid (P = 0.027), lipid (P = 0.0063), and phthiocerol biosynthetic processes (P = 0.0027) such as *fadB2*, *fadD26*, *pks1*, *pks15*, and *ppsA-C*, which are all upregulated (Fig 5C–5D) [38–41]. We also find upregulation of *icl* which encodes an isocitrate lyase (Fig 5C–5D); Serafini *et al.* found similar transcriptional changes in Mtb grown on lactate and also showed that *icl* is essential for Mtb growth in lactate [42]. In addition to *icl*, this gene set is enriched with other genes implicated in oxidation-reduction processes like electron transport components *nuoD* and *nuoE*, which are both downregulated in the strain carrying the evolved allele (Fig 5C–5D) [43].

Type VII secretion components (P = 0.032) are also enriched in this gene set. We find that the *espACD* operon is downregulated in Mtb with the evolved -18 G>T allele compared to the -18 T>G ancestral allele in lactate-containing media (Fig 5C–5D). This operon is essential for secretion of the virulence effectors EsxA and EsxB via the ESX-1 Type VII secretion system [44,45]. Serafini *et al.* also observed downregulation of *espACD* expression in H37Rv (which expresses the derived *lldD2* –18 G>T allele) grown on lactate compared to glucose [42]. Other genes involved in pathogenesis (P = 0.029) are upregulated including those encoding PE13, T-cell receptors specific to which are associated with TB disease control [46], and PPE18, which is a component of the M72/AS01E TB vaccine candidate [47]. These data indicate that lactate metabolism is not just feeding anaplerotic needs but is a host-relevant signal that the bacterium responds to by changing expression of key virulence systems.

## Discussion

Previous work determined that *lldD2* is undergoing host-derived positive selection and identified lineage and sub-lineage defining mutations in *lldD2* [8,10,28]. We expanded this analysis to illustrate the stepwise addition of *lldD2* mutations that occurred over evolutionary history and show this augments Mtb replicative fitness during lactate growth. These phenotypic data provide experimental evidence that *lldD2* diversification is functional and beneficial to Mtb, and that the prevalence of *lldD2* variation is not solely due to genetic drift. More complex genetic interactions may also be possible such that *lldD2* variants are especially important in the setting of other metabolic or virulence pathway mutations.

A very simple model for the selective advantage conferred by *lldD2* evolution is that lactate is an abundant carbon source in some host environments, and Mtb is simply tuning lactate fitness via the evolution of *lldD2* to increase anabolic capacity. However, our data support and

expand a more complex model proposed by Pethe *et al.* in which a detoxification pathway exists between lactate and methylglyoxal, a toxic byproduct produced spontaneously from DHAP and G3P during glycolysis and lipid metabolism [48]. Methylglyoxal reactivity is known to mediate some of the harmful effects of sugar phosphate accumulation [34]. Consistent with this model, we find that DHAP and G3P accumulate in the *ΔlldD2* strain when grown in the presence of lactate and glycerol. Interestingly, LDH knockout or deficiency has also been associated with an increase in G3P in *Drosophila melanogaster* larvae, cancer cell lines, and human patients [49–52]. The *ΔlldD2* strain exhibits significant growth impairment in lactate media, even with glycerol present. *LldD2* gain-of-function variants mitigate this accumulation This suggests to us that detoxification may also be a significant selective pressure in addition to the anaplerotic importance of LldD2.

Transcriptomic analysis demonstrates that *lldD2* mutations have effects beyond lactate anaplerosis. Evolutionary tuning of lactate metabolism affects genes involved in phthiocerol synthesis–implicated in macrophage evasion and immunopathology [53]–as well as ESX1 function. This suggests that Mtb modifies its virulence strategies when exposed to lactate and that the evolution of *lldD2* could be driven in part by cell wall and/or virulence remodeling.

Lactate production is induced by infected macrophages after stimulation with IFNγ [25], which suggests lactate is an immune-regulated carbon substrate. Mtb may utilize lactate as a signal of the host immune state and adjust growth and virulence accordingly. This would suggest that lactate is an ancient selective pressure that is still relevant today as *lldD2* diversification is currently ongoing [8]. Assuming that lactate has always been a relevant feature of the host environment, it is unclear why optimized *lldD2* function did not evolve long ago. This could simply reflect the stochasticity of evolution. The Mtb strain carrying the most ancestral allele of *lldD2* is significantly impaired during growth in lactate, which may suggest that LldD2 once mediated enzyme activity distinct from lactate oxidation. Over time, stochastic acquisition of mutations that enabled lactate oxidation may have conferred the ability of LldD2 to function as an LDH, and these mutations were selected for their pleiotropic effects on Mtb fitness during lactate growth. There is precedence for the hypothesis that LldD2 was derived from an enzyme with a different function; a previous study demonstrated that SNPs can confer LDH activity to a bacterial glycerol dehydrogenase [54]. Alternatively, the availability of human lactate during infection could be dynamic, which would force Mtb to continually adapt. This change in host environment could be shaped by shifts in immune landscapes, perhaps from BCG vaccination, altered nutritional state, or altered transmission pressures–modulating either lactate exposure or the optimal response to this host cue. The dynamics of human lactate availability may also be shaped by variations in genes that govern the complexity of human lactate metabolism [55–57]. Future work may consider if human genetic backgrounds demonstrate variation in genes that regulate this process, as this may affect the fixation of *lldD2* mutations in different Mtb populations over time.

In all, we provide experimental evidence that ancient lineage and sub-lineage defining mutations and more contemporary homoplastic mutations in *lldD2* reflect stepwise gain-of-function. This work has provided novel insight into Mtb adaption to host stress, and as a result, elucidated the intersection of metabolic remodeling and virulence.

## Materials and methods

### *lldD2* mutant strain construction

We completed RecET oligo-mediated recombineering according to the protocol as described [8,58] in the Mtb H37Rv reference strain to introduce the V3I and V263M mutations into the

native locus of the coding region of *lldD2* and to revert the -18 G>T mutation acquired by H37Rv to the ancestral allele. We completed Sanger sequencing to screen the transformants for acquisition of these mutations. We retained strains that failed to incur *lldD2* mutations for use as H37Rv controls.

*ΔlldD2* was constructed in H37Rv according to the methodology detailed by Griffin *et al.* [59]. We then complemented *ΔlldD2* with a series of different *lldD2* sequences that recapitulate the evolutionary history of the gene. We also complemented *ΔlldD2* with the first 300 base pairs upstream of *lldD2*, this null sequence serves as a negative control. Complementation was accomplished by transformation with a plasmid carrying a kanamycin resistance cassette and an *lldD2* sequence of interest that integrates at the L5 phage site. *LldD2* was expressed under the direction of the pUV15-TetON promoter [60]; because the plasmid we used lacks the repressor TetR, *lldD2* is constitutively expressed.

For Fig 4E, we also cured the hygromycin cassette from the *lldD2* locus of the *ΔlldD2* strain [59]. The cassette was removed by transforming *ΔlldD2* with a plasmid constitutively expressing a Cre recombinase that executes excision between the loxP sites. As an empty vector control, the H37Rv strain used to construct *ΔlldD2* was transformed with the same plasmid. We completed Sanger sequencing of the transformants to confirm loss of the hygromycin cassette and tested for susceptibly to hygromycin by inoculating the strains in 10 mL of 7H9 media (7H9 salts with 0.2% glycerol, 10% OADC supplement, 0.05% Tween80) and 50 μg/mL of hygromycin and using a spectrophotometer to monitor growth via $OD_{600}$ for seven days.

The Msm strains expressing *lldD2* were constructed in the reference strain mc$^2$155. We transformed mc$^2$155 with a Giles integrating plasmid carrying a nourseothricin resistance cassette, a sequence corresponding to 200 base pairs upstream of *lldD2*, and the *lldD2* sequences of interest encoding the first 60 amino acids of LldD2 with the C-terminus fused to Renilla luciferase. With the inclusion of the upstream region, *lldD2* expression is driven by the native promoter. The plasmid is modified from Rock *et al.* [61]. See S1 Table for a list of all strains used in the study.

All transformations were recovered on 7H10 solid media (7H10 salts with 0.2% glycerol, 10% OADC supplement, and the appropriate antibiotic). The integrase gene remained in the recombination vector for all constructs.

## Mtb clinical strains

The clinical strains utilized in this study are part of a larger collection of 1,635 samples originally isolated from patient sputum in Ho Chi Minh City, Vietnam and published by Holt *et al.* [30]. Our group transformed a subset of these strains with an L5-integrating plasmid carrying a kanamycin resistance cassette and a unique genetic barcode for a separate analysis [62,63]. These barcoded clinical strains were used to conduct the growth curves in the current analysis. S1 Table lists the *lldD2* genotype and accession numbers for these strains.

## Phylogenetic tree and *lldD2* mutation annotation

The phylogenetic tree depicting the global diversity of MTBC strains was adapted from the figure originally published by Liu *et al.* [29]. The phylogenic tree was designed using iTOL (version 6.6) [64]. The *lldD2* SNPs were called in reference to the inferred ancestral genome of the MTBC most recent common ancestor [65]. These mutations were utilized to annotate the LldD2 diagram using the lollipops package (v1.6.0) [66]. The *lldD1* and *lldD2* mutational burden figures in Fig 2B–2D were based on the sequences from the ~50,000 clinical isolates analyzed by Liu *et al.* [8].

## Growth curves

The growth curves with the recombinant Mtb strains and the clinical isolates were completed by diluting mid-log phase cultures (grown in 7H9 at 37°C with shaking) to an $OD_{600}$ of 0.005 in 10 mL of the indicated media that included 50 μg/mL hygromycin, 20 μg/mL kanamycin, and/or 20 μg/mL zeocin. 7H12 consists of 7H9 salts, 0.1% casamino acids, and 0.05% tyloxapol. Carbon sources include 0.2% w/v lactate, 0.2% v/v glycerol, or 0.1% w/v lactate and 0.1% v/v glycerol (0.2% lactate + glycerol). Each strain was inoculated in three separate inkwells per media condition to serve as replicates. We measured the $OD_{600}$ at the indicated time points after inoculation. We performed growth curves independently twice for each set of strains.

## *lldD2* expression analyses

We completed qPCR to measure *lldD2* expression in the point mutant strains. To do this, we inoculated each strain into 10 mL of 7H9 + 50 μg/mL hygromycin and allowed them to grow at 37°C with shaking to mid-log phase. In triplicate, we diluted each strain into 10 mL of 7H9 so they would reach an $OD_{600}$ of 0.5 after 3 days of additional growth. Afterward, we spun down the cultures, resuspended in 1 mL of TRIzol (Invitrogen, Waltham, MA, USA), and completed bead beating to lyse the cells. We added 30% total volume of chloroform then performed RNA extraction and cleanup using the RNA Clean & Concentrator kit (Zymo Research, Irvine, CA, USA). We obtained cDNA using random hexamers with a SuperScript IV reverse transcriptase kit (Thermo Fisher Scientific, Waltham, MA, USA). We then performed qPCR to quantify transcript abundance. Expression of *lldD2* was normalized to *sigA* expression. We utilized the following primers: *sigA* 5′-CAAGTTCTCCACCTACGCTAC-3′ and 5′-GTTGATCACCTCGACCATGT-3′; *lldD2* 5′- CCGCGACATCGAGTTTCACCCG-3′ and 5′- GGATGGACATCCCGTTGCGGACATC-3′. We completed three independent rounds of culturing, RNA extraction, and qPCR.

## Renilla luciferase assay

To quantify LldD2 production, we performed a Renilla luciferase assay using the Msm LldD2-Renilla luciferase fusion constructs [61]. Each Msm strain was cultured to mid-log phase in 5 mL of 7H9 + ADC (7H9 salts with 0.2% glycerol, 10% ADC supplement, 0.05% Tween80) plus 25 μg/mL nourseothricin at 37°C with shaking. In triplicate, the cultures were then back diluted into 7 mL of 7H9 + ADC and 25 μg/mL nourseothricin such that they would reach log phase after 24 hours of growth. We measured the $OD_{600}$ of the cultures and harvested four $OD_{600}$ units of cells by pelleting the samples. After removing the supernatant, we completed the assay according to protocol (Renilla Luciferase Assay System, Promega, Madison, WI, USA). We performed the assay in 96-well Costar white plates (Corning, Corning, NY, USA) and quantified luciferase fluorescence with the VarioSkan Flash plate reader (Thermo Fisher Scientific). The Renilla assay was performed independently four times.

## Alamar blue assays

Clofazimine (CFZ) and trifluoperazine sensitivity were assessed with an Alamar blue reduction assay. Strains were grown to mid-log phase in 7H9 and then diluted to an $OD_{600}$ of 0.003 in the indicated media. The assay was performed as described previously but without shaking [12]. For the 7H12–0.2% L-lactate comparisons, reduction was measured at $OD_{570}$ with a microplate reader. After correcting for the background $OD_{570}$ readings utilizing the no-Mtb wells, percent growth was quantified by normalizing the readings for the strain and drug conditions by the readings for Mtb-only wells [12]. We also completed Alamar blue assays with

7H12–0.2% L-lactate + glycerol to control for different growth rates in media with lactate as the sole carbon source. For these assessments, the minimum inhibitory concentration (MIC) was determined as the lowest concentration of antibiotic that inhibited the Alamar blue reagent from transitioning from blue to purple or pink after three days of incubation. We performed the CFZ MIC assay three independent times and the trifluoperazine assay two independent times; we observed similar results as demonstrated by the 7H12–0.2% L-lactate experiments.

## Lactate metabolic flux assay

We cultured the point mutant strains and the cured *ΔlldD2* strains to mid-log phase in 7H9 with the appropriate antibiotics, then back-diluted the strains so they would read an $OD_{600}$ of ~1 two days later at 37˚ C with shaking. We collected the cells via vacuum filtration by applying 1 mL of culture to a 0.22 μm mixed cellulose filter membrane (MilliporeSigma, St. Louis, MO, USA) affixed to a filter membrane holder (MilliporeSigma). We collected three samples per strain. The Mtb-laden filters were placed on modified 7H10 solid media (7H10 salts, 0.1% casamino acids, 0.1% v/v glycerol, 0.1% w/v L-lactate, and the appropriate antibiotic) and incubated for 7 days at 37˚ C to allow for biomass formation. We then pulsed with $^{13}$C-labelled lactate by transferring the Mtb biomass filters onto 7H10 solid media with 0.1% v/v glycerol, 0.1% w/v $^{13}$C-labeled lactate, and the appropriate antibiotic for 8 hours. Afterward, we quenched metabolism by placing each filter into a 40:40:20 solution of acetonitrile:methanol:water chilled with dry ice for 1 minute. The transfer buffer, Mtb biomass, and filter were then transferred into bead-beating tubes filled with 0.1 mm Zirconia/silica beads for metabolite extraction. We completed bead-beating 6 times for 30 seconds at 6 m/s with intermittent cooling down for 1 minute on ice. After centrifugation, we double-filtered the samples with Spin-X centrifuge 0.22 μm cellulose-acetate filter tubes (Corning) for removal out of the BSL3 laboratory. We used the Pierce BCA protein assay kit (Thermo Fisher Scientific) to measure residual protein in the samples in order to normalize by cell abundance in downstream analyses. Samples were stored at -80˚ C.

Metabolite extract samples were analyzed using a quadrupole-orbitrap mass spectrometer coupled with hydrophilic interaction chromatography (HILIC) as the chromatographic technique. Chromatographic separation was achieved on an XBridge BEH Amide XP Column (2.5 μm, 2.1 mm × 150 mm) with a guard column (2.5 μm, 2.1 mm X 5 mm) (Waters, Milford, MA, USA). For the gradient, mobile phase A was water:acetonitrile 95:5, and mobile phase B was water:acetonitrile 20:80, both phases containing 10 mM ammonium acetate and 10 mM ammonium hydroxide. The linear elution gradient was: 0 ~ 3 min, 100% B; 3.2 ~ 6.2 min, 90% B; 6.5. ~ 10.5 min, 80% B; 10.7 ~ 13.5 min, 70% B; 13.7 ~ 16 min, 45% B; and 16.5 ~ 22 min, 100% B, with a flow rate of 0.3 mL/ min. 5 μL of samples were injected using the autosampler at 4˚C. Needle wash was applied between samples using methanol:acetonitrile:water at 40:40:20. The mass spectrometer used was Q Exactive HF (Thermo Fisher Scientific), which scanned from 70 to 1000 *m/z* with switching polarity. The resolution was 120,000. Metabolites were identified based on accurate mass and retention time using EI-Maven (Elucidata, Cambridge, MA, USA) with an in-house library. Correction of isotope-labeled lactate or $^{13}$C-natural abundance was performed in R using the package AccuCor [67]. We calculated the relative abundance of isotopically labeled metabolite species by dividing the integrated peak area of each isotope species by the summed peak area of all labeled species.

## Differential expression analysis

In order to perform RNA-seq, we back-diluted six mid-log phase cultures of each strain so they would reach an $OD_{600}$ of 0.5 after two days of growth in 10 mL of 7H9 media with antibiotic at 37˚ C with shaking. We pelleted each culture, removed the supernatant, resuspended

the cells, and inoculated three cultures per strain with 7H9 and 50 μg/mL hygromycin or 7H12 + 0.2% w/v L-lactate and antibiotic. We allowed these cultures to shake at 37° C for 6 hours, then we extracted and cleaned the RNA as described above. We used the KAPA RiboErase kit (Roche, Basel, Switzerland) with Mtb custom rRNA targeting oligos for rRNA depletion. To prepare the RNA sequencing libraries, we followed the manufacturer's instructions for the KAPA RNA HyperPrep kit (Roche). We used 25 ng of input RNA. One 7H9–18 T>G sample was lost during the process. The prepared RNA libraries were sequenced with the MiSeq Reagent Kit v3 (150-cycle, Illumina, San Diego, CA) with a 75bp paired-ended setup. Sequencing reads were mapped to the H37Rv reference genome with the bwa mem pipeline [12,68]. We used the ht-seq tool 'count' to assign genomic features to the aligned reads; reads that aligned to rRNA were removed [12,69]. To perform the differential expression analysis, we used DESeq2 according to the standard parameters [70].

## Supporting information

**S1 Fig. Growth curves of L2 Mtb clinical isolates with the indicated *lldD2* alleles.** Each dot denotes the average of three technical replicates, the bars represent the standard deviation. The p-value indicates the results of ordinary-one way ANOVA tests conducted at each time point for each media condition, comparing the $OD_{600}$ of the ancestral and variant strains. Sidak's multiple comparison correction was performed.
(PDF)

**S2 Fig.** Comparison of the area under the curve for the growth curves shown in Fig 3D (A), Fig 3F (B), and Fig 4D (C). Three replicates are shown, error bars indicate the standard deviation. P-values indicate the results of an ordinary one-way ANOVA with Dunnett's multiple comparison test. Representative of two independent experiments.
(PDF)

**S3 Fig. Alamar blue assay of trifluoperazine.** P-value indicates the results of two-way ANOVA with Dunnett's multiple test correction. Triplicate replicates shown, error bars represent the standard deviation.
(PDF)

**S4 Fig.** (A) Growth curves of the *lldD2* allelic variants with glycerol as the sole carbon source. All cultures started at $OD_{600}$ 0.005 at day 0. Triplicate replicates shown, error bars represent the standard deviation. Representative of two independent experiments. (B) Area under the curve analysis of the growth curves shown in (A). Three replicates are shown, error bars indicate the standard deviation. P-values indicate the results of an ordinary one-way ANOVA with Dunnett's multiple comparison test.
(PDF)

**S5 Fig. Growth curves and corresponding area under the curve analysis for the strains utilized for the metabolic flux assay shown in Fig 4E.** All cultures started at $OD_{600}$ 0.005 at day 0. Triplicate replicates shown, error bars represent the standard deviation. H37Rv carries the -18 G>T *lldD2* mutation. Kan[R] refers to kanamycin resistance. P-values indicate the results of unpaired t-tests. Representative of two independent experiments.
(PDF)

**S6 Fig. [13]C-lactate metabolic flux analysis results.** The error bars represent the standard deviation of three technical replicates. Blue indicates labeled carbon ions, grey represents total carbon ions.
(PDF)

**S1 Table. Mtb clinical and recombinant strain list.**
(XLSX)

**S2 Table. Metabolite counts from the $^{13}$C-lactate flux assay.**
(XLSX)

**S3 Table. Mtb gene expression data during 7H9 and 7H12–0.2% L-lactate growth.**
(XLSX)

**S1 Data. S1 Data includes data underlying all figures except Fig 4F (S2 Table) and Fig 5B–5C (S3 Table).**
(XLSX)

## Author Contributions

**Conceptualization:** Sydney Stanley, Xin Wang, Qingyun Liu, Sarah M. Fortune.

**Formal analysis:** Sydney Stanley, Xin Wang, Qingyun Liu, Young Yon Kwon, Abigail M. Frey.

**Funding acquisition:** Sheng Hui, Sarah M. Fortune.

**Investigation:** Sydney Stanley, Young Yon Kwon, Abigail M. Frey, Nathan D. Hicks, Andrew J. Vickers.

**Methodology:** Sydney Stanley, Xin Wang, Qingyun Liu, Young Yon Kwon, Nathan D. Hicks.

**Resources:** Sheng Hui, Sarah M. Fortune.

**Supervision:** Sheng Hui, Sarah M. Fortune.

**Visualization:** Sydney Stanley, Xin Wang, Qingyun Liu, Abigail M. Frey.

**Writing – original draft:** Sydney Stanley.

**Writing – review & editing:** Sydney Stanley, Xin Wang, Qingyun Liu, Sarah M. Fortune.

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
