## [Decision Letter · Decision Letter 0]

17 Dec 2023

Dear Dr. Fortune,

Thank you very much for submitting your manuscript "Ongoing evolution of the Mycobacterium tuberculosis lactate dehydrogenase reveals the pleiotropic effects of bacterial adaption to host pressure" for consideration at PLOS Pathogens. As with all papers reviewed by the journal, your manuscript was reviewed by members of the editorial board and by several independent reviewers. The reviewers appreciated the attention to an important topic. Based on the reviews, we are likely to accept this manuscript for publication, providing that you modify the manuscript according to the review recommendations.

Most of the concerns of reviewer 2 can likely be addressed in the rebuttal letter.

Sincerely,

Helena Ingrid Boshoff

Academic Editor

PLOS Pathogens

Debra Bessen

Section Editor

PLOS Pathogens

Kasturi Haldar

Editor-in-Chief

PLOS Pathogens

orcid.org/0000-0001-5065-158X

Michael Malim

Editor-in-Chief

PLOS Pathogens

orcid.org/0000-0002-7699-2064

Most of the concerns of reviewer 2 can likely be addressed in the rebuttal letter.

Reviewer Comments (if any, and for reference):

Reviewer's Responses to Questions

**Part I - Summary**

Reviewer #1: The work is largely well performed, analyzed and interpreted and delivers novel insights into the evolution of metabolic adaptation of Mycobacterium tuberculosis. The work focuses on lactate metabolism and the lldD2 gene, which encodes a quinone-dependent lactate dehydrogenase. To me, the greatest contribution of this manuscript is the discovery of another enzyme or pathway to catabolize lactate. This might well be another lactate dehydrogenase, that has not been yet characterized, yet, it is novel, and provides some evidence of redundancy, which should agree with the overall hypothesis put forth by the authors, that lactate is an important carbon source for M. tuberculosis.

Strengths:

Carefully done phylogenetic and evolutionary analysis implicating LldD2 and lactate metabolism, as point of evolution in modern mycobacterial strains.

Strong data on genetic manipulation of mycobacteria, demonstrating that lldD2 polymorphisms are indeed affecting lactate metabolism.

RNA-seq analysis reveals genetic programs that have already been implicated in lactate metabolism, directly or indirectly, and therefore provides further evidence of the broad cellular effects imparted by a metabolic defect in lactate metabolism.

Weaknesses.

Clofazimine MIC determination should have been carried out in medium without glycerol. And a 2-fold change in MIC is not proof of change (that is noise). The minimum acceptable change in such an assay would be 4-fold. Therefore, I would remove this data from the paper, as it is "negative". Not really supporting the hypothesis, which is otherwise sound.

Physiology and metabolism results are largely good and definitive. I just did not understand why the authors jumped on "methylglyoxal" as the explanation for the phenotypes observed. The authors presented no proof that methylglyoxal is formed or that accumulates. This is similar to what was conveyed in the Pethe study, without any proof. The authors must remove any mention of methylglyoxal of the manuscript. Or show that it is formed and accumulates.

Reviewer #2: This manuscript argues that the gene lldD2 (Rv1872c) which encodes a lactate dehydrogenase, has been positively selected for in the human host. Clearly, Mtb has continued to evolve, with selective pressures, but the authors claim that BCG vaccination may have played a role in acquiring these mutations, but no animal experiments have been included to test their hypothesis. As an alternative to host immune selective pressure, the authors have not described that lactate dehydrogenase converts pyruvate to lactate with the concomitant conversion of NADH to NAD+. Since resistance to Isoniazid and Ethionamide can be mediated by alterations in the NADH/NAD ratios, could not these drugs play a role in the evolution of the mutations in lldD2? These possibilities are not considered. Also, the use of 7H12 for some of the assays is not explained and this could be playing a role in the mutations examined in their analyses. Further clarifications will be needed.

Reviewer #3: The manuscript presents a detailed analysis of genetic variation of the lldD2 gene in Mycoabcterium tuberculosis. First by assessing natural variation in clinical strains and then by introducing the variants by allelic replacement and assessing the phenotype. In general the manuscript is very well written. To the best of my knowledge in silico and in vitro analysis are nicely done and show a very nice example of the continuous adaptation of the bacteria to host challenges.

**Part II – Major Issues: Key Experiments Required for Acceptance**

Reviewer #1: The authors need to re-do their CLF MIC determination in medium without glycerol, having lactate as sole carbon source. Changes in growth rate in the mutants should not be a problem.

The authors must remove any mention that methyglyoxal is toxic to Mtb and differentially toxic under their experimental conditions as no direct proof is provided of increase methylglyoxal formation or accumulation.

Reviewer #2: Specific questions to the authors:

1. In the list of 296 strains, were any of them drug resistant?

2. You make a point that 99.6% of the 59,00 strains analyzed have mutations in lldD2 gene compared to the 10% in the lldD1 gene. Are there no other genes of the 4000 genes common to the 50,000 strains that possess this amount of divergence? Is this only found in lldD2?

3. For the experiments with the 10 clinical isolates with lldD2 mutations, what are the genotypes for other drug resistances?

4. You note that the strains are grown on 7H12 with lactate but give no explanation as to why you switched to 7H12. Is this because the strains fail to grow on 7H9 with lactate? 7H12 has casein and I would hypothesize that lldD2 mutants could grow on 7H12 with lactate, but not on 7H9 with lactate because of the alteration of the NADH/NAD+ ratios in the mutants. Such a phenotype has been observed in M. smegmatis mutants that were selected to be resistant to Isoniazid and ethionamide when selected on Mueller Hinton agar. The mutantations map to ndh and the strains have altered NADH/NAD ratios which caused serine/glycine auxotrophies by inactivating the first step of serine biosynthesis (Miesel, L et al. “NADH dehydrogenase defects confer isoniazid resistance and conditional lethality in Mycobacterium smegmatis.” Journal of bacteriology vol. 180,9 (1998): 2459-67. doi:10.1128/JB.180.9.2459-2467.1998). Could the alterations in lactate dehydrogenase activities be causing the strains to acquire auxotrophic phenotypes?

5. It would have been best to make the mutants in the isogenic strains in the clinical isolates from Vietnam. Also, transformants were made using an L5 integration vector with a kanamycin-resistance gene in a vector that has yet to be published. Is this a derivative from the Lee et al paper or the Stover et al, paper and was the integrase gene still in the vector. What was the value of the bar codes/ Were the transformants selected on 7H12? If selected on 7H9, did you select for lldD2 reversions?

6. The problem with the H37Rv constructs are the notorious problem of PDIM loss when culturing H37Rv in Middlebrook 7H9. The Berney lab has a recent paper, but unless you genome sequenced the strains after even a simple transformation, you could be selecting for PDIM. I suspect that the RNAseq data showing overexpression of PDIM genes is not due to lldD2h mutations but rather the acquisition of PFIM mutations. Unfortunately, this makes it impossible to know if the differences in transcription result from lldD2 mutations alone.

7. While clofazimine has been suggested to alter electron flow, the Nuo complex is not essential for the growth of Mtb in vitro or in mice. This makes the clofazimine argument challenging to interpret.

8. The possibility of the in vivo selection for lldD2 mutations in humans is very intriguing. However, just growth enhancement in vitro is not sufficient evidence for this hypothesis to be justified. Do you have any evidence that lldD2 mutations cause a survival advantage that prevents killing? Most all drug resistance mutations provide a resistance to a killing assault.

9. The argument that lldD2 is modified significantly more than the other 4000 genes of Mtb would be an interesting paper, but has this not been argued in the cited papers?

Reviewer #3: Regarding natural variation. The narrative in the introduction suggests that what we are looking to a process of positive selection. While this is likely true for many variants, and particularly homoplastic or other recently arisen, some others, particularly at the root of the tree may have fixed by other processes related to drift (bottlenecks, founder effects….etc). Another possibility is that some changes are compensatory to others (early changes are now detrimental so strains are trying to compensate the fitness cost). Somewhat this is discuss later including the fact that probably the pathogen is facing a dynamic fitness landscape what leads to my second comment

The suggestion is that adaptation is probably linked to lactate accumulation in the macrophages. Is there any information about how lactate levels may change during infection or even in healthy conditions? Are also reported differences between human genetic backgrounds that may explain the early fixation of mutations you see in the phylogeny?

Finally, in terms of analysis, the suggestion is that lldD2 mutations may help the bacteria to better survive host stresses which in turn may help the bacteria to spread between host. While transmission is a complex phenotype I am wondering if there is some hint in your dataset (probably from Vietnam?) that strains carrying the mutation are more likely to be in cluster (or have shorted branch length as a surrogate)

**Part III – Minor Issues: Editorial and Data Presentation Modifications**

Reviewer #1: No minor issues.

Reviewer #2: More details on the genetic manipulations should be included. It is unclear how the mutations in the lldD2 were selected for after recombineering.

Reviewer #3: (No Response)

PLOS authors have the option to publish the peer review history of their article (what does this mean?). If published, this will include your full peer review and any attached files.

Reviewer #1: No

Reviewer #2: **Yes: **William R Jacobs

Reviewer #3: No

Figure Files:

Data Requirements:

Reproducibility:

References:

---

## [Editor Report · Decision Letter 1]

13 Feb 2024

Dear Dr. Fortune,

We are pleased to inform you that your manuscript 'Ongoing evolution of the Mycobacterium tuberculosis lactate dehydrogenase reveals the pleiotropic effects of bacterial adaption to host pressure' has been provisionally accepted for publication in PLOS Pathogens.

Best regards,

Debra E Bessen

Section Editor

PLOS Pathogens

Debra Bessen

Section Editor

PLOS Pathogens

Michael Malim

Editor-in-Chief

PLOS Pathogens

orcid.org/0000-0002-7699-2064
---

## [Editor Report · Acceptance letter]

25 Feb 2024

Dear Dr. Fortune,

We are delighted to inform you that your manuscript, "Ongoing evolution of the Mycobacterium tuberculosis lactate dehydrogenase reveals the pleiotropic effects of bacterial adaption to host pressure," has been formally accepted for publication in PLOS Pathogens.

Best regards,

Michael Malim

Editor-in-Chief

PLOS Pathogens

orcid.org/0000-0002-7699-2064